# Xylem Sap Bleeding as a Physiological Indicator in Grapevine: Genotype and Climate Influence

**DOI:** 10.3390/plants14172807

**Published:** 2025-09-08

**Authors:** Eleonora Nistor, Alin Dobrei, Andreea Dragoescu-Petrica, Eleonora Cataldo, Florin Sala, Gabriel Ciorica, Alina Georgeta Dobrei

**Affiliations:** 1Department of Horticulture, Faculty of Engineering and Applied Technologies, University of Life Sciences “King Mihai I” from Timisoara, 119 Calea Aradului, 300645 Timisoara, Romania; nisnoranisnora@gmail.com (E.N.); alin1969tmro@yahoo.com (A.D.); 2Department of Agricultural Technologies, Faculty of Agriculture, University of Life Sciences “King Mihai I” from Timisoara, 119 Calea Aradului, 300645 Timisoara, Romania; andreeadragoescu@yahoo.com; 3Department of Agriculture, Food, Environment and Forestry Sciences and Technologies (DAGRI-UniFi), Florence University, 50100 Florence, Italy; eleonora.cataldo@unifi.it; 4Department of Soil Sciences, Faculty of Agriculture, University of Life Sciences “King Mihai I” from Timisoara, 119 Calea Aradului, 300645 Timisoara, Romania; florin_sala@yahoo.com; 5Department 6 Cardiology, Internal Medicine Ambulatory Discipline, “Victor Babes” University of Medicine and Pharmacy, Piata Eftimie Murgu 2, 300041 Timisoara, Romania

**Keywords:** bleeding, cultivar, macronutrients, micronutrients, phenols, sap, xylem

## Abstract

The aim of the research was to investigate several xylem sap parameters (onset, sap bleeding duration and intensity, and main chemical components) in four cultivars (‘Cabernet Sauvignon’, ‘Merlot’, ‘Muscat Ottonel’, and ‘Pinot Noir’) grown in the climate of western Romania over three consecutive growing seasons (2022–2024). Understanding early-season sap characteristics is relevant for optimizing vineyard management and improving grape output in fluctuating environmental conditions. Sap onset and duration differed significantly among cultivars and years (*p* < 0.05), with warmer springs resulting in earlier and longer sap bleeding. ‘Pinot Noir’ consistently exhibited the highest and earliest xylem sap flow (8.2–10.8 mL/vine/day). Chemical profiling revealed cultivar- and year-dependent variation in soluble solids, macro- and micronutrients, proteins, organic acids, and phenolic compounds. ‘Muscat Ottonel’ and ‘Pinot Noir’ had higher phenolic contents, while ‘Merlot’ displayed lower metabolic activity inferred from sap composition. Years with higher precipitation showed reduced phenolic acid and resveratrol concentrations. Principal component analysis highlighted strong effects of both cultivar and season on sap chemistry, with warmer years favoring nutrient- and metabolite-rich profiles, particularly in ‘Pinot Noir’ and ‘Muscat Ottonel’. These findings confirm that xylem sap bleeding is a sensitive indicator of grapevine reactivation, shaped by genotype and climate. Monitoring sap traits at dormancy release can serve as an early diagnostic tool to guide pruning, irrigation, and fertilization and supports the selection of climate-resilient cultivars and rootstock–scion combinations.

## 1. Introduction

Grapevine sap bleeding—also known as xylem exudation or guttation—is the natural physiological process in which water is exuded from the plant; typically in liquid form; due to root pressure. This phenomenon occurs in vineyards every spring after early pruning, usually before budburst, once the temperature in the top 25 cm of the soil exceeds 6 °C, signaling the end of winter dormancy [1]. As the weather warms, the vine’s roots begin to absorb water and nutrients from the soil and transport them to all plant organs [2]. Grapevine bleeding is common in both one-year-old and perennial vine wood. Factors such as cultivar, climate, and soil moisture influence the rate, volume, onset, and composition of the sap, making it a useful indirect indicator of soil-root interaction, vine health, and vitality at the start of the new growing season [3]. Restoring vascular function following dormancy is critical for sap bleeding, because during dormancy, xylem vessels may have embolisms or air obstructions [4]. The sap rehydrates the vessels, dissolves trapped gases, and restores hydraulic continuity—the uninterrupted flow of water through the plant—during bud expansion and leaf growth; thereby enabling transpiration and photosynthesis to proceed [5].

Sap contains a complex mixture of mineral nutrients, mainly nitrogen (N) compounds, potassium (K), calcium (Ca), and magnesium (Mg), along with organic molecules such as amino acids, sugars, and phytohormones [6]. These compounds are transferred from the roots and storage tissues to the buds and cambial tissues, facilitating metabolic and meristematic activity [7]. These variations are primarily controlled by vine vigor, genetic, and physiological factors [8], as well as by interactions between rootstock genotype and grapevine age [9]. In viticulture, rootstock selection is critical because it influences several physiological functions, including nutrient absorption, drought tolerance, and key phenological processes such as budding and blooming [10]. A limitation of the experimental design is that the cultivars used in this study were grafted onto different rootstocks. Consequently, the observed variations in exudation and sap composition may reflect both graft-specific characteristics and rootstock effects.

Among the cultivars, ‘Muscat Ottonel’, which matures more rapidly, has a much faster sap flow but for a notably shorter duration than ‘Cabernet Sauvignon’, which has a later onset and a moderate sap flow rate [11]; however, its exudation tends to last longer [12]. Due to strong root pressure and excellent compatibility with the rootstock, ‘Merlot’ vines often produce a large volume of sap [13]. Depending on soil structure and nutrient availability, ‘Pinot Noir’ exhibits deficiencies in magnesium and potassium uptake [14]. The rootstock influences hydraulic conductance and nutrient uptake efficiency, which ultimately affect the sap composition [3]. Genetic variability, such as clonal diversity, can result in varying sap bleeding patterns within cultivar groups [15]. Cultivars with high metabolic rates or well-developed root systems tend to produce sap rich in nitrogen compounds and phytohormones, which influence bud and shoot development [16].

The bleeding process in grapevines is highly sensitive to environmental conditions, which can influence the frequency, duration, and composition of sap exudation [17]. Key environmental factors include soil and air temperature, soil moisture, the timing of early spring pruning, and winter onset [18]. Temperature plays a critical role in both initiating and sustaining sap flow. The optimal temperature range for sap bleeding to begin varies depending on the location and cultivar [19]. Extended cold periods or irregular fluctuations between warm days and cold nights may slow or disrupt this process. Frost events in early spring, following a warm spell, can temporarily block sap flow, negatively impacting bud development and vine recovery [20]. Soil moisture influences root pressure and sap flow, whereas drought conditions are associated with reduced bleeding or even its cessation [10], particularly in older vines or those with deeper root systems [21].

Pruning time significantly affects both the intensity and timing of sap bleeding [19]. Pruning in late winter or early spring, when root pressure is rising, can result in substantial sap leakage. In contrast, early pruning during dormancy may reduce sap flow due to delayed physiological reactivation [11]. The volume of exuded sap is also influenced by the number of pruning wounds and the diameter of the cuts [22].

In vineyard management, monitoring sap variability is essential because it provides valuable information about grapevine health, physiology, nutrient availability, and root activity. This information supports informed decisions on fertilization, irrigation, and the timing and methods of pruning [23]. Understanding sap bleeding enables growers to adjust pruning techniques to reduce vine stress and enhance wound healing [24]. Furthermore, deviations from typical sap bleeding patterns—such as reduced or delayed sap flow—may indicate abiotic stress factors (e.g., frost damage, drought, or soil compaction) or serve as early signs of vascular disease that require prompt attention [25].

This knowledge can facilitate the development of management strategies tailored to grapevine physiology and environmental conditions, especially as climate variability increases. Such strategies can improve vine resilience, reduce resource waste, and enhance long-term vineyard performance. Compared to well-studied viticultural regions in Western Europe, Australia, and North America, sap bleeding remains under-researched in Eastern Europe. This has created a significant gap in the literature and a strong need for regionally relevant data—particularly in countries like Romania; where diverse temperature zones and traditional viticultural practices intersect.

This study aims to address a gap in knowledge by analyzing and comparing the xylem sap characteristics of four wine grape cultivars grown in western Romania: ‘Cabernet Sauvignon’, ‘Merlot’, ‘Muscat Ottonel’, and ‘Pinot Noir’. Specifically, it focuses on how cultivar and climate influence sap exudation patterns and nutritional content during early spring sap flow. We hypothesize that sap volume and composition will vary among cultivars, reflecting differences in root pressure, metabolic activity, and nutrient transport efficiency. Therefore, this study also aims to serve as a baseline for future studies and provide practical insights for vineyard management in similar agro-climatic environments.

## 2. Results

### 2.1. Climatic Conditions During the Study (2022–2024)

Grapevine development is strongly influenced by climate—especially temperature and precipitation—at each developmental stage. Climate data analyzed over the last three growing seasons (2022–2024) reveal considerable variability (Figure 1). In January 2023, the weather was warm, with an average minimum temperature (MTmin) of 1.58 °C, and in March it reached 1.74 °C, conditions that favored root activity and sap bleeding. Similarly, in early spring 2024 (March), temperatures continued to rise (MTmin: 2.85 °C), leading to an advanced growing season, early bud burst, and sap bleeding. In contrast, the colder March of 2022 (MTmin: −2.29 °C) delayed sap bleeding and bud burst. This variability highlights the grapevine’s sensitivity to even slight temperature fluctuations during dormancy. Overall, the earlier spring warming in the 2023 and 2024 growing seasons advanced sap bleeding and bud burst by approximately one to two weeks compared to 2022.

As temperatures rose in April and May, the weather became favorable for bud burst, shoot growth, and inflorescence development. Spring warming in 2024—with average temperatures of 13.93 °C in April and 17.82 °C in May—was clearly evident compared to previous years (2022 and 2023); indicating earlier and accelerated vegetative growth. Precipitation during these months was variable: in May 2023, 82.4 mm of rainfall supported early growth, whereas April and May 2024 were drier, resulting in low humidity stress associated with limited nutrient availability.

The flowering and berry set stages, in late May and early June, were strongly influenced by both temperature and precipitation. Across years, the average temperature in June exceeded 20 °C, which favored timely flowering. However, precipitation varied significantly: June 2023 received 93.5 mm of rain, whereas June 2024 had only 41.3 mm. This contrast had important implications for fruit set success, as wet weather can increase the risk of diseases—particularly downy mildew.

July and August represent the peak of the growing season, which is critical for berry development and the onset of veraison. These summer months were very hot in all three years of study; however, 2024 was exceptionally hot and dry. In July, the average temperature exceeded 26.27 °C, with a maximum average temperature (MTmax) reaching 34 °C. August followed a similar pattern, with an average temperature of 25.88 °C and an MTmax of 34.11 °C. The hot weather accelerated ripening but also increased the proportion of dehydrated and sunburned berries, particularly due to the lack of precipitation (only 5.9 mm in August 2024, strongly contrasting with the 106.9 mm recorded in August 2023).

Harvest time in September 2024 was characterized by warm temperatures (mean temperature of 19.6 °C) and unusually high precipitation for the past two decades (103.1 mm), which adversely affected the harvest and increased the risk of berry rot. In contrast, October 2022 and 2024 were cooler and drier, with average temperatures below 13.03 °C and precipitation under 15 mm, conditions that favored the harvest of late-ripening cultivars. Comparatively, 2023 featured optimal harvest conditions, with moderate temperatures and rainfall.

Finally, the transition to dormancy in November and December began earlier in 2024, as temperatures declined more rapidly than in previous years. The minimum average temperature (MTmin) in November fell to −0.69 °C in 2024, compared to 2.67 °C in 2023, indicating a shorter growing season. This earlier cooling influenced the post-harvest nutrient assimilation period and pruning schedules.

### 2.2. Xylem Sap Exudation Main Parameters

#### 2.2.1. The Sap Bleeding Onset

The onset of sap bleeding varied significantly among the four cultivars and across the three growing seasons (2022–2024), reflecting the influence of climate variability, mainly the lowest temperature in early spring. In March 2022, the minimum temperature reached −2.29 °C, which delayed the physiological reactivation of vine roots. Consequently, sap bleeding began later, with the first exudates recorded between March 14 and 20, depending on the cultivar. In contrast, the warmer March temperatures in 2023 ((mean minimum temperature (MTmin): 1.74 °C)) and especially in 2024 (MTmin: 2.85 °C) promoted earlier sap bleeding, observed as early as February 26 in some cultivars (Table 1). Among the cultivars, ‘Pinot Noir’ showed the earliest onset of bleeding, occurring 2 to 4 days before the other cultivars in all three seasons; ‘Muscat Ottonel’ followed, while ‘Cabernet Sauvignon’ and ‘Merlot’ were the latest, especially in 2022. This variation may be attributed to cultivar genotype, which influences earlier root activity and xylem pressure after pruning.

#### 2.2.2. Bleeding Duration

Bleeding duration also varied significantly, ranging from 10 to 21 days. In the 2022 growing season, the bleeding duration was shorter due to slower physiological reactivation and faster budburst once temperature thresholds were exceeded (Table 2).

During the 2023 and 2024 growing seasons, the bleeding duration was notably longer, particularly for the ‘Muscat Ottonel’ and ‘Pinot Noir’, both cultivars displaying extended sap flow during the pre-bud burst stage. In contrast, the ‘Merlot’ cultivar constantly showed a shorter bleeding duration (especially in 2022), possibly due to the lower root pressure and a delayed reaction to pruning. The results reveal a seasonal pattern, characterized by an earlier onset and prolonged bleeding duration during the warmer late winter and early spring. These findings support the hypothesis that root pressure and sap flow are sensitive to temperature and that each cultivar exhibits distinct physiological behavior during dormancy and early seasonal reactivation.

ANOVA results show a statistically significant difference (*p* < 0.001) in sap bleeding duration across all cultivars and growing seasons. Sap bleeding duration increased significantly between 2022 and 2024 in all cultivars, indicating that this consistent upward trend may be influenced by climate and cane maturation.

#### 2.2.3. Sap Bleeding Intensity

Sap bleeding intensity (mL/vine/day) varied across growing seasons (2022–2024) and cultivars. This variation was influenced by both the physiological characteristics of each cultivar and the environmental conditions of each season (Table 3). Across all three growing seasons, ‘Pinot Noir’ registered the highest bleeding intensity consistently, reaching a moderate peak sap flow (10.8 ± 1.2 mL/vine/day) in 2024, a year characterized by a warm early spring and higher humidity. ‘Muscat Ottonel’ also demonstrated a high sap flow, but lower than that of ‘Pinot Noir’; its bleeding intensity remained relatively stable across seasons, with the highest values recorded in 2023 and 2024.

In contrast, ‘Merlot’ and ‘Cabernet Sauvignon’ consistently exhibited lower bleeding intensity, often below 6 mL/vine/day. The lowest bleeding intensity was recorded in ‘Merlot’ during the 2022 season, which correlated with the delayed sap onset caused by low temperatures in March (MTmin: −2.29 °C).

Both cultivars exhibited improved sap bleeding in 2024; however, their values remained below those of ‘Pinot Noir’ and ‘Muscat Ottonel’.

The highest intensity of sap exudation was recorded in 2024, correlating with an early onset of exudation and a longer bleeding duration. This trend is also associated with higher temperatures and increased root pressure, suggesting that early spring temperatures significantly influence the vigor and amplitude of sap exudation in grapevine cultivars. These findings confirm that bleeding intensity is not completely dependent on the cultivar but is also highly sensitive to environmental dynamics, making it a reliable physiological marker for dormancy breaking and grapevine reactivation.

In Cabernet Sauvignon, the increasing volume of bleeding sap from one year to the next suggests a strong cultivar response, enhancing its ripening potential and suitability for producing high-quality wine.

The *p*-values for ‘Merlot’ and ‘Muscat Ottonel’ indicate highly significant changes, reflecting increased physiological activity each year. The lowest *p*-value observed in ‘Pinot Noir’ suggests that the high rate of sap bleeding in this cultivar is primarily attributable to strong hydraulic activity and vigor during the early growing season.

### 2.3. Sap Chemical Composition (2022–2024)

Key parameters analyzed across all four cultivars during the 2022–2024 growing seasons included total soluble solids (TSS), sap pH, and electrical conductivity (EC). Each parameter provided insights into the vine’s physiological status and potential stress indicators.

#### 2.3.1. Total Soluble Solids (TSS)

Total soluble solids measured in Brix (°Brix) exhibited a significant variability between cultivars and growing seasons (Table 3). ‘Muscat Ottonel’ constantly registered the highest TSS values, indicating a high accumulation of sugars or osmotic concentration in the sap. In contrast, the lowest values were observed in ‘Merlot’ during the 2022 growing season, reflecting the cultivar’s physiological response to early-season conditions.

ANOVA *p*-values (*p* < 0.05) for ‘Cabernet Sauvignon’, ‘Merlot’, and ‘Muscat Ottonel’ indicate statistically significant differences in TSS across the 2022–2024 growing seasons. This suggests that environmental conditions and physiological changes in the grapevines significantly influenced TSS accumulation in these cultivars.

In contrast, ‘Pinot Noir’ had a *p*-value slightly above the conventional significance threshold, indicating a trend toward variation over time, but without strong statistical evidence to support a consistent seasonal effect on TSS in this cultivar.

#### 2.3.2. Sap pH

Sap pH values remained relatively stable across cultivars and growing seasons, ranging from 6.1 to 6.9 (Table 3). Statistical analysis revealed no significant differences between cultivars (*p* > 0.05), indicating that sap pH is a relatively consistent trait under the experimental conditions of this study.

#### 2.3.3. Electrical Conductivity (EC)

Most values for the four cultivars exhibited low (0.43 mS/cm) to moderate (0.63 mS/cm) electrical conductivity (EC). For ‘Cabernet Sauvignon’, EC values increased notably in 2023 compared to 2022, followed by a gradual decrease in 2024 (Table 3).

This variability was statistically significant, suggesting that climate fluctuations and physiological responses influenced the sap’s ionic composition. In the ‘Merlot’ cultivar, differences in electrical conductivity (EC) were also statistically significant, indicating sensitivity to seasonal climate variations and physiological changes in the grapevine.

For ‘Muscat Ottonel’, EC remained relatively stable across seasons. Despite the narrow range of fluctuation, the change was statistically significant, although with a smaller amplitude compared to ‘Cabernet Sauvignon’ and ‘Merlot’. Similarly, ‘Pinot Noir’ showed only minor seasonal variations. The ANOVA *p*-value, slightly above the conventional threshold of significance, indicates that seasonal growing conditions did not have a statistically significant effect on EC in this cultivar.

#### 2.3.4. Macronutrient Composition

ANOVA statistical analysis revealed significant differences (*p* < 0.05) across growing seasons for all macronutrients in each cultivar (Table 4). This indicates that seasonal factors—including soil nutrient availability; weather conditions; and grapevine physiological stages—substantially influenced macronutrient levels during the early growing season.

‘Muscat Ottonel’ exhibited the highest levels of nitrogen (N), potassium (K), calcium (Ca), and sulfur (S), especially in 2024, indicating a pattern of nutrient remobilization. ‘Merlot’ and ‘Cabernet Sauvignon’ follow similar trends, but at lower levels. ‘Pinot Noir’ also presented a significant increase, especially in N and P, with statistical significance (all values *p* ≤ 0.031).

These results support the hypothesis that xylem sap composition during bleeding reflects nutrient dynamics, cultivar specificity, and environmental responsiveness, which may provide valuable information and strategies for fertilization and vineyard management during the early growing stages.

#### 2.3.5. Micronutrient Composition

The analysis of micronutrients in xylem sap across the 2022–2024 growing seasons revealed temporally consistent and statistically significant trends for all six microelements measured in each cultivar (Table 5). The concentration of iron (Fe) in ‘Cabernet Sauvignon’ increased in 2023, followed by a slight decrease in 2024.

A similar moderate increase was observed for Mn, Zn, Cu, B, and Mo. All ANOVA *p*-values were below 0.05, indicating significant year-to-year variability. This variability reflects seasonal physiological changes and environmental factors that influence the mobilization of micronutrients.

Micronutrient levels in the sap of ‘Merlot’ exhibited a similar trend to that observed in ‘Cabernet Sauvignon’, with a notable increase in Mn and Zn over the three-year period. Iron and boron levels increased steadily, while Cu and Mo showed smaller yet statistically significant increases. All *p*-values were below 0.031, confirming a consistent temporal effect on micronutrient availability and uptake.

In 2024, the sap of ‘Muscat Ottonel’ exhibited the highest concentrations of Fe and Zn. The progressive increase in most micronutrients from 2022 to 2024 was statistically significant, suggesting either a strong physiological demand or efficient nutrient mobilization during early grapevine development. In ‘Pinot Noir’, a significant annual increase was observed for all micronutrients, with Zn and B reaching their highest levels in 2024. ANOVA results highlighted a strong effect of the growing season on sap micronutrient composition, although inter-annual variability was lower compared to that of ‘Muscat Ottonel’. These findings indicate the combined influence of genotype and environmental factors on the micronutrient content of xylem sap during grapevine reactivation.

#### 2.3.6. Protein Content

In all cultivars, total protein content increased significantly from one growing season to the next (*p* < 0.05), indicating enhanced metabolic activity during sap bleeding as grapevines prepare for bud burst (Table 6). The highest protein levels were observed in the sap of ‘Pinot Noir’, whereas ‘Merlot’ exhibited the lowest values. These differences reflect cultivar-specific physiological traits related to xylem loading and protein mobilization during early growth.

#### 2.3.7. Peroxidase Activity

Peroxidase activity increased significantly in all cultivars across the studied growing seasons, with *p*-values below 0.05, confirming the influence of seasonal factors on enzyme activity (Table 6). This increasing trend reflects a more intense oxidative metabolism and physiological preparation for bud swelling and shoot emergence.

#### 2.3.8. Polyphenol Oxidase Activity

Polyphenol oxidase activity exhibited a pattern similar to that of peroxidase, with a significant increase observed in all cultivars (*p* < 0.05) (Table 6).

This increase corresponds with the anticipated rise in phenolic metabolism, which is associated with vascular activity and early stress signaling during the initial stages of grapevine development at the beginning of the growing season.

#### 2.3.9. β-Glucosidase Activity

β-glucosidase, an enzyme involved in glycoside hydrolysis and secondary metabolite activity, also showed a significant increase over the years in all cultivars (*p* < 0.05) (Table 6). This activity may be linked to the mobilization of aroma precursors and to the production of defense-related compounds as grapevine physiology advances toward active growth.

#### 2.3.10. Organic Acid Content in Xylem Sap

Organic acids are essential components of xylem sap, serving as intermediates in respiratory pathways, osmoregulators, and chelating agents—chemical compounds that bind metal ions to form stable; water-soluble complexes—for nutrient transport.

The seasonal dynamics of organic acids provide valuable insights into early vine metabolism, particularly during the sap bleeding stage when root pressure facilitates the upward translocation of metabolites. In this study, the concentrations of the four major organic acids—malic acid; tartaric acid; citric acid; and oxalic acid—were measured in the xylem sap of ‘Cabernet Sauvignon’; ‘Merlot’; ‘Muscat Ottonel’; and ‘Pinot Noir’ cultivars (Table 7).

In ‘Cabernet Sauvignon’, a consistent increase in malic and tartaric acid concentrations was observed over the three growing seasons. This trend is associated with increased root metabolic activity and earlier mobilization of stored organic acids in response to rising spring temperatures and improved vine vigor. The increase in citric and oxalic acid concentrations in 2024 indicates increased cycle turnover and detoxification processes within the tricarboxylic acid (TCA) cycle, which are associated with greater nutrient mobilization. Significant ANOVA *p*-values (all <0.05) confirm that year-to-year variations are statistically significant and influenced by both environmental dynamics and grapevine age progression. Merlot exhibited a similar upward trend for all organic acids, although with lower concentrations compared to ‘Cabernet Sauvignon’. Malic acid increased steadily across the years, reaching 78.0 mg/L in 2024, reflecting consistent metabolic reactivation in early spring.

Tartaric and citric acids increased moderately, possibly due to enhanced phloem-xylem exchange and contributions from root tissues. Oxalic acid levels gradually increased, reaching 10.2 mg/L in 2024, suggesting its role in ion chelation and calcium balance homeostasis. These trends are consistent with the intermediate phenological development of ‘Merlot’ and the balanced root pressure during sap flow.

Organic acids affect how nutrients reach roots. These effects match changes in macronutrient levels. Higher organic acid levels boost the mobility and solubility of micronutrients. This aids certain cultivars in absorbing more nutrients. Thus, sap composition shows a mix of genetic traits, nutrient flow, and yearly weather. All ANOVA *p*-values were < 0.05, confirming the significance of yearly variation.

#### 2.3.11. Bleeding Sap Phenolic Compound Analysis (2022–2024)

Key phenolic compound concentrations were measured in xylem sap during the early bleeding stage (Table 8). The early-developing and metabolically active cultivar, ‘Muscat Ottonel’ presented the highest levels of all organic acids among the four cultivars. In 2024, malic acid reached a peak of 94.4 mg/L, while citric acid increased to 27.1 mg/L. This elevated organic acid profile is likely associated with the development of the aromatic profile and early xylem activity, which mobilizes stored acids to support early vegetative growth. The increase in oxalic acid (from 10.1 to 12.0 mg/L) may reflect intense detoxification of free calcium and pH modulation during the rapid environmental changes in spring. Significant ANOVA results confirm strong seasonal effects, probably amplified by the early phenology and higher metabolic demand of this cultivar.

‘Pinot Noir’ also exhibited a significant increase in organic acid concentration, particularly malic and citric acids. These results align with the known sensitivity of ‘Pinot Noir’ to environmental factors and its rapid development during spring. A moderate increase in tartaric acid (from 48.8 to 53.4 mg/L) suggests ongoing remobilization from root and trunk storage tissues, while the accumulation of oxalic acid likely supports cation balance and osmotic adjustment.

The narrow standard deviations indicate a consistent physiological response across years, supported by significant ANOVA *p*-values. These patterns highlight the physiological contrast of ‘Pinot Noir’ compared to more vigorous cultivars such as ‘Cabernet Sauvignon’.

‘Cabernet Sauvignon’ demonstrated a progressive increase in all measured phenolic compounds across the years. Resveratrol levels were significantly increased, indicating enhanced stilbene synthesis. Flavonoids, catechins, and phenolic acids followed a similar upward trend, with statistically significant year-to-year differences (*p* < 0.05). This suggests an increase in the biosynthesis of phenolic metabolites during sap bleeding, primarily driven by environmental factors. A similar pattern was observed in ‘Merlot’ sap, with consistent increases in phenolic compounds. Resveratrol, catechins, and flavonoids showed significant increases over the years. Both resveratrol and phenolic acids also increased, but to a lesser extent. These trends suggest active phenolic transport or metabolic activity during the transition from dormancy to the growth phase.

The ‘Muscat Ottonel’ cultivar exhibited high levels of resveratrol—likely due to its genetic predisposition; early and active metabolism; and efficient xylem function—as well as elevated catechin content. A consistent increase in flavonoids and folic acid further supports the strong phenolic profile of this cultivar. These trends highlight the intrinsic capacity of ‘Muscat Ottonel’ for early phenol synthesis during sap bleeding.

The highest phenolic acid content was found in ‘Pinot Noir’. The increases in resveratrol and flavonoid levels were statistically significant, indicating a strong antioxidant potential of sap during early phenological stages.

One-way ANOVA revealed significant differences (*p* < 0.05) in the level of all phenolic compounds among cultivars across years. This indicates that the accumulation of phenolic compounds during sap bleeding was significantly influenced by climate variability, grapevine physiology, and environmental conditions in each growing season.

### 2.4. Principal Component Analysis

Principal Component Analysis of sap macronutrients and micronutrients. According to the PCA biplot (Figure 2), the first two principal components (F1 and F2) account for 83.10% and 9.37%, respectively, of the total variance. Most nutrients—including potassium (K); boron (B); manganese (Mn); phosphorus (P); zinc (Zn); calcium (Ca); magnesium (Mg); sulfur (S); and nitrogen (N)—show strong positive association. In contrast, iron (Fe) is negatively correlated with molybdenum (Mo).

Sap samples from ‘Muscat Ottonel’ (2023–2024) and ‘Cabernet Sauvignon’ (2024) contained higher levels of most macro- and micronutrients and lower levels of Fe and Cu. Conversely, ‘Merlot’, ‘Cabernet Sauvignon’, and ‘Pinot Noir’ samples collected in 2022 exhibited higher Fe and Cu concentrations and lower levels of other macro- and micronutrients.

The second PCA biplot (Figure 3) shows that the first two principal components (F1 and F2) together explain 98.46% of the total variance in the dataset, with F1 accounting for 91.71% and F2 for 6.75%, indicating an excellent representation of the original data in the biplot. The F1 axis differentiated the samples mainly on the basis of organic acid and phenolic compound concentrations, with strong positive loadings observed for both.

The ‘Pinot Noir’ sap samples from the 2023 and 2024 growing seasons presented high levels of analyzed organic acids and phenolic compounds. In contrast, sap samples collected in 2022 from ‘Merlot’, ‘Cabernet Sauvignon’, and ‘Muscat Ottonel’ were associated with lower concentrations of organic acids and polyphenols.

The PCA biplot analysis indicates that sap from ‘Pinot Noir’ collected each year reflects rich profiles of organic acids and phenolic compounds, whereas sap collected in 2022 from ‘Merlot’ and ‘Cabernet Sauvignon’ showed lower metabolite levels. These distribution patterns illustrate the significant influence of both the cultivar and the growing season on the biochemical composition of the sap.

The PCA illustrates the interdependent accumulation pattern of organic acids and polyphenols in xylem sap and demonstrates the utility of multivariate analysis in identifying the unique metabolic profiles of different cultivars. These findings provide valuable insights for understanding grapevine physiology and for selecting genotypes with desirable profiles for healthy vines, good grape quality, and potential wine-making performance.

## 3. Discussion

The present study provides conclusive evidence that the patterns of sap parameters and chemical composition are significantly modulated by both the cultivar genotype and the climatic variation of the growing season, particularly during the critical stages of dormancy and reactivation in early spring. Each cultivar in this study was grafted onto a different commercial rootstock, using common viticultural practices in the region. It is already known that rootstocks influence various aspects of grapevine physiology, including hydraulic conductivity, nutrient uptake, and the timing of phenological stages, such as bud burst and sap flow rate. The differences observed among cultivars in bleeding intensity, soluble solids, and enzymatic activity may result from rootstock-graft interaction, rather than solely from the graft genotype. Because rootstock treatments were not included in the experimental design, their effects could not be statistically distinguished from those of the cultivar.

### 3.1. Environmental Influence on Phenology and Bleeding Intensity

The results confirmed that the onset of sap flow and the bleeding duration are highly sensitive to temperature dynamics in early spring. Warmer growing seasons in 2023 and 2024 showed earlier and longer sap bleeding. These findings align with previous research, which suggests that temperature and root activity are key factors in sap bleeding onset [26,27]. Zheng et al. (2020) [1] noted that sap bleeding in grapevines is affected by cultivar genotype, irrigation level, temperature, pruning system, and environmental stressors like frost and drought. Notably, the ‘Pinot Noir’ cultivar consistently bled earlier and more intensely, likely due to a faster developmental profile and higher hydraulic conductivity, possibly as a consequence of improved xylem vessel functionality [28]. In contrast, ‘Cabernet Sauvignon’ and ‘Merlot’ exhibited later sap bleeding onset and lower intensity. This is likely related to slower root reactivation and specific xylem dynamics in those genotypes.

Significant differences were observed in sap bleeding duration among cultivars, with ‘Muscat Otonel’ and ‘Pinot Noir’ showing longer durations in 2023 and 2024, and ‘Merlot’ having the shortest duration. These results agree with Grall et al.’s (2005) [29] findings that sap flow is influenced by vine physiology and grape cultivar genetics, including vine age and rootstock type. ‘Pinot Noir’ consistently had the highest sap flow intensity, while ‘Cabernet Sauvignon’ and ‘Merlot’ had significantly lower flows over three years, especially in the cooler year of 2022. This is consistent with Bouamama-Gzara et al. (2022) [3] detection of significant variations in xylem sap flow in five Tunisian grape varieties across several years. Our study results also show that sap flow intensity is influenced by both cultivar physiology and annual changes in external conditions, such as humidity and temperature.

### 3.2. Sap Composition—Physiological Readiness and Environment Modulation

Total soluble solids (TSS), electrical conductivity (EC), and pH reflect the grapevine’s metabolic readiness during xylem sap bleeding [30]. ‘Muscat Ottonel’ showed high TSS values, suggesting more intense osmotic activity and sugar mobilization in this earlier-developing cultivar. The pH remained stable across cultivars and seasons. This supports previous findings that pH is buffered in xylem sap during dormancy [31,32]. EC fluctuations, especially in ‘Cabernet Sauvignon’ and ‘Merlot’, reflect seasonal changes in ion mobilization and nutrient uptake, consistent with observations in comparable perennial species [33]. Variability of sap composition varies by cultivar and growing season, such as higher total soluble solids (TSS) content in ‘Muscat Ottonel’ or lower in ‘Merlot’ and variable electrical conductivity (EC) and relative pH stability, which are consistent with the results of studies conducted on the ‘Muscadine’ variety by Andersen and Brodbeck (1989) [34], who demonstrated that both sap flow and solute flux increase with temperature and depending on the cultivar.

### 3.3. Macronutrient and Micronutrient Profile—Indicators of Cultivar-Specific Mobilization

Data showed that ‘Cabernet Sauvignon’ and ‘Muscat Ottonel’ reached higher concentrations of macro- and micronutrients across years, supporting the idea that nutrient remobilization from perennial tissues is coordinated with increased xylem pressure [35]. Micronutrients such as Zn, Mn, and Fe show significant upward trends (*p* < 0.05), indicating intensified root-to-shoot translocation during sap bleeding [36]. The rich micronutrient profile of ‘Muscat Ottonel’ likely results from its higher metabolic demands during early development. The variation in macronutrient concentrations (N, P, Ca, S) between cultivars and seasons, particularly in ‘Muscat Ottonel’ in 2024, reflects a physiological dynamic previously observed by Harris et al. (2022) [37] in the French-American hybrid grapevine “Chambourcin”, where factors like year and phenological stage significantly influence nutrient composition, and the rootstock genotype interacts with environmental conditions for nutrient absorption and translocation. The observed trends in sap micronutrient composition, with increases in Fe and Zn concentrations in ‘Muscat Ottonel’ and ‘Pinot Noir’ and in Mn, Cu, B, and Mo over time, are consistent with previous research, such as Ahmed et al. (2024) [38] finding that micronutrient levels in grapevines vary seasonally due to environmental conditions and cultivar-specific physiological requirements.

### 3.4. Organic Acids and Phenolic Compounds—Indicators of Grapevine Vigor and Metabolism

A key finding is the strong correlation between organic acids and phenolic compounds, as demonstrated by PCA biplot analysis. Organic acids (malic, tartaric, and citric) play vital roles in respiratory metabolism and nutrient chelation, while phenolic compounds (resveratrol, catechins, and flavonoids) serve as antioxidants and stress-response metabolites [1,16]. The parallel increase of these compounds, especially in ‘Pinot Noir’ and ‘Muscat Ottonel’, indicates a coordinated response to environmental conditions and developmental cues. This confirms the idea that early xylem sap composition is not just a passive transport fluid but a dynamic medium reflecting active physiological adjustment in response to environmental factors and phenological stages [39,40]. ‘Pinot Noir’ exhibited the most diverse organic acid and polyphenol profiles, likely due to its known high metabolic sensitivity and low buffering capacity, making it particularly responsive to environmental changes [37]. In contrast, ‘Cabernet Sauvignon’ and ‘Merlot’ sap from 2022 exhibited lower levels of organic acids and phenolic compounds, which corresponded with cooler temperatures and delayed physiological activation. The rising trends of resveratrol and peroxidase activity support the hypothesis of enhanced oxidative signaling and defense mechanisms during the transition out of dormancy. The activities of β-glucosidase and polyphenol oxidase further illustrate how sap biochemistry reflects readiness for growth and defense—findings that align with studies in other perennial systems; such as olive trees [41]. During the initial stages of sap bleeding, all four cultivars show significant increases in organic acids and phenolic compounds that vary by cultivar and growing season. This is supported by Zheng et al. (2020) [1] research on xylem sap from the ‘Rosario Bianco’ cultivar, which confirmed the presence and quantification of these organic acids along with various phenolic compounds, including flavonoids and polyphenols. This highlights the sap’s diverse composition and its functional roles in grapevine metabolism and stress defense.

### 3.5. Principal Component Analysis (PCA)

PCA results supported the strong correlation between organic acid and phenolic compound accumulation, confirming that these metabolites reflect common metabolic pathways activated during sap bleeding. This suggests that the biochemical profile of xylem sap can reliably indicate grapevine physiological status. The high F1 loadings for samples from 2023 to 2024, especially in ‘Muscat Ottonel’ and ‘Pinot Noir’, show that environmental factors drive metabolic increases, consistent with literature on the impact of warming trends on grapevine metabolism [42,43]. The PCA results are consistent with the observations of Teixeira et al. (2023) [44] on phloem sap of Portuguese grapevine cultivars affected by *Flavescence dorée*, where PCA differentiated healthy sap samples from infected ones, with the first two components explaining 94% of the variance. This similarity in findings confirms PCA as a powerful tool for disentangling compositional dynamics between cultivar and growing conditions.

## 4. Materials and Methods

### 4.1. Study Area Description

The research was carried out at the University of Life Sciences “King Mihai I”, located at the northern edge of Timișoara, in western Romania. The experimental plots are situated at 45.7896° N latitude and 21.2149° E longitude, with an elevation of 91 m above sea level. This area is characterized by a temperate climate with Mediterranean influences, providing favorable conditions for grapevine cultivation. The vineyard features a collection of cultivars.

The experimental plots were established on flat terrain, offering uniform exposure to temperature and precipitation. The east-to-west orientation of the vine rows provides consistent sunlight exposure and air movement within the canopy, promoting healthy grapevine development and physiological responses, including sap bleeding.

The area soil type is chernozem with a calcium subtype, which can affect grapevine health and grape composition. The soil texture ranges from loam to clay loam, balancing water retention and aeration. The upper soil layer (0–40 cm deep) is rich in organic matter, with a granular to sub-angular structure that facilitates root development and microbial activity. Below this layer, the soil is more compact but still accessible to deeper roots, especially in older vines.

Soil pH ranges from 6.5 to 7.4, indicating neutral to slightly alkaline conditions. This pH range is favorable for the availability of essential macro- and micronutrients, including potassium, phosphorus, and magnesium. The presence of calcium carbonate in deeper soil layers improves berry skin quality and grape composition. The presence of calcium carbonate in deeper soil layers improves berry skin quality and grape composition.

### 4.2. Climate Data

Climate data over three years (2022–2024), including minimum, maximum, and daily average temperatures, together with the total monthly precipitation, were acquired from the Timisoara Meteorological Station. The key phenological stages of the grapevine growth season—dormancy ending; sap bleeding; bud burst; blooming; berry development; harvesting; and dormancy onset—were then examined in connection with these data; which had been processed into monthly averages and totals (for precipitation). Climate variability across years was then compared to assess its influence on grapevine development and xylem sap dynamics.

### 4.3. Experimental Design

A complete randomized block design (RCBD) was used to account for trial research, taking into account the variability in vineyard soil and microclimate. For each cultivar, 10 vines per plot were selected, with three replications (blocks) randomly distributed across the research field, resulting in 30 vines per cultivar. Winter pruning was performed in the first ten days of March (2022 and 2023) and late February (2024), before bud burst. To minimize variations among cultivars due to vine canopy structure or bud load, 2–3 buds were retained on each cordon, with evenly spaced spurs.

### 4.4. Plant Material

The research focused on four widely cultivated grapevine cultivars (*Vitis vinifera* L.) in the temperate viticultural areas: ‘Cabernet Sauvignon’, ‘Muscat Ottonel’, ‘Merlot’, and ‘Pinot Noir’. The vines were between 10 and 15 years old, with mature and stable grape yields, making them suitable for physiological and soil-plant interactions evaluations. The cultivars were grafted onto rootstock selected for compatibility with location conditions: ‘SO_4_′ (Selection Oppenheim 4) for ‘Cabernet Sauvignon’ and ‘Merlot’, ‘1103 Paulsen’ for ‘Muscat Ottonel’ (drought tolerance and soil rich in lime), and ‘101-14 Mgt’ (Millardet et de Grasset) for ‘Pinot Noir’ (chosen for moderate vigor and climate adaptability). These combinations followed regional viticultural recommendations for optimal vine health and fruit quality.

The grapevine rows were planted at 2.5 interrows and 1.2 m between vines within rows, providing a density of about 3.333 vines/hectare. The vine rows were oriented from east to west. All vines were trained using a double Guyot vertical (VSP) trellis system to support the canopy and manage microclimate. Grapevine management, including pruning and soil maintenance, was standardized to minimize variability among cultivars. No chemical treatments or irrigation were applied during the research.

#### Sap Samples Collection

Sap samples were collected before bud burst every year from 2022 to 2024. We monitored sap flow from March to early April, when soil temperature and root pressure trigger sap exudation from the xylem. To avoid variability in sap flow, canes were pruned 2 cm above the internodes in all cultivars. After pruning, a plastic tube (50 mL) was securely attached to the cane tip and sealed tightly to prevent evaporation losses. Sap flow was measured every two days from one cane per vine. After each measurement, the collection system was removed, the cane was re-pruned, and the tube was re-sealed until the next measurement. All measurements were taken in the morning (08:00–10:00) to minimize the influence of temperature fluctuations on exudation rates.

### 4.5. Measurement Parameters

For sap flow dynamics in each cultivar, the following parameters were recorded: bleeding intensity, onset, and duration of bleeding. Bleeding intensity was defined as the sap volume per vine over 24 h, expressed in milliliters (mL/vine/day). On each sampling date, sap volume was measured individually for all vines, and the daily sap bleeding rate was averaged per cultivar and replication. Onset of bleeding was recorded as the first day after winter pruning when the first sap drops appeared, indicating physiological reactivation of root pressure. Bleeding duration was defined as the total consecutive number of days (or intervals) with measurable sap flow (>0.5 mL/vine/day) before bleeding ceased at bud burst.

*Sap chemical composition* was analyzed to evaluate the nutritional and physiological profile. On a set date, one sap sample from each replication (cultivar × block) was collected (*n* = 3 per cultivar per year, corresponding to one composite sample per block) and transported in a cool bag to the laboratory for analysis. Chemical analyses included the determination of total soluble solids (TSS) in °Brix, using a digital handheld refractometer (Atago PAL-1, 501 Omaeda, Fukaya-shi, Saitama 369-1246, Japan) to estimate the sugar concentration. Electrical conductivity (EC) and pH were measured with a multiparameter device (Hanna Instruments HI9811, Hanna Instruments Service S.R.L., Cluj Napoca, Romania) to assess sap acidity and ionic strength, respectively. Mineral nutrient concentrations—including nitrogen (N); calcium (Ca); magnesium (Mg); potassium (K); phosphorus (P); iron (Fe); zinc (Zn); sulfur (S); boron (B); molybdenum (Mo); and copper (Cu)—were quantified by ion chromatography (IC) using a Dionex ICS-5000 system.

The organic sap composition was evaluated, focusing on organic acids (malic, tartaric, citric, and oxalic acids) and phenolic compounds (flavonoids, phenolic acids, resveratrol, and catechins), determined by high-performance liquid chromatography (HPLC) and HPLC with diode-array detection (DAD), respectively, following standard extraction and derivatization procedures. Protein content in the xylem sap was measured in triplicate using the Bradford assay. Sap samples were centrifuged to remove debris, and 50 µL of the supernatant was mixed with 1 mL of Bradford reagent. After 10 min of incubation, absorbance was measured at 595 nm using a spectrophotometer (Shimadzu UV-1800 UV-Vis, Shimadzu Corp., Kyoto, Japan). During sap sample collection, daily minimum and maximum temperatures and rainfall were recorded using an automated weather station located less than 1 km from the vineyard.

#### Analytical Chemistry

To ensure transparency and reproducibility of the analytical procedures, the parameters for the chemical analyses (IC, HPLC, and HPLC-DAD) are presented in Table 9. The table includes details on mobile phases, gradients, detection wavelengths, calibration intervals, limits of detection and quantification (LOD/LOQ), quality control (QC) procedures, and the Bradford assay for protein, summarized for completeness.

### 4.6. Statistical Analysis

All field and laboratory data were compiled and organized using Microsoft Excel (Microsoft Office 365; May 2021, Version 2105 (Build 14026.20246) Windows 11) and GraphPad Prism (version 9.0, GraphPad Software, San Diego, CA). A one-way analysis of variance (ANOVA) evaluated differences in sap bleeding intensity, sap composition, and duration among cultivars. When results showed significant differences (*p* < 0.05), Tukey’s test was applied as a post-hoc multiple comparison procedure to identify statistically distinct group means. Descriptive statistics, including means and standard deviations, were calculated for each cultivar and variable. Principal component analysis (PCA) assessed multivariate relationships among the four cultivars and identified patterns of similarity or dissimilarity. Statistical significance was determined at the 95% confidence level (α = 0.05).

### 4.7. Limitations

While the research design was chosen for viability across different growing seasons and several wine grape cultivars, some limitations must be acknowledged. Natural variability in grapevine vigor (influenced by rootstock performance and previous pruning practices) could influence sap flow and composition. Furthermore, weather data accuracy from a nearby automatic weather station may not perfectly reflect the real microclimate from the vineyard. To minimize these potential sources of errors, all vines from the research field were pruned on the same day each season, maintaining consistent crop loading and cutting position. Sap was collected during the same day interval (08:00–10:00) to avoid variations in root pressure. Canes of identical diameter and sampling instruments were used for sap induction. Laboratory analyses followed standard protocols to ensure data accuracy and minimize variability. These controls ensured that most of the observed differences may be attributed to the cultivar characteristics or environmental influence, rather than to potential procedural inconsistencies.

## 5. Conclusions

Xylem sap bleeding represents a physiological process influenced by both cultivar genotype and climate variability, reflecting metabolic activity at the onset of the growing season. Across three years (2022–2024), the onset, duration, and intensity of sap flow varied significantly among cultivars. ‘Pinot Noir’ and ‘Muscat Ottonel’ exhibited the strongest and earliest sap bleeding, stimulated by warmer conditions. In contrast, ‘Merlot’ and ‘Cabernet Sauvignon’ showed weaker and more delayed sap flow responses, particularly during the last two growing seasons. Sap composition differed across cultivars and years, with ‘Pinot Noir’ and ‘Muscat Ottonel’ enriched in phenolic and organic compounds. Principal Component Analysis confirmed that both vintage and cultivar influenced sap biochemical profiles. Precipitation also played a decisive role: higher rainfall diluted phenolic acids and resveratrol, whereas drier seasons favored polyphenol accumulation.

These findings indicate xylem sap bleeding as a potential diagnostic tool for vineyard management and an early-season physiological indicator of vine performance, supporting decision-making in precision viticulture and climate change adaptation.

## Figures and Tables

**Figure 1 plants-14-02807-f001:**
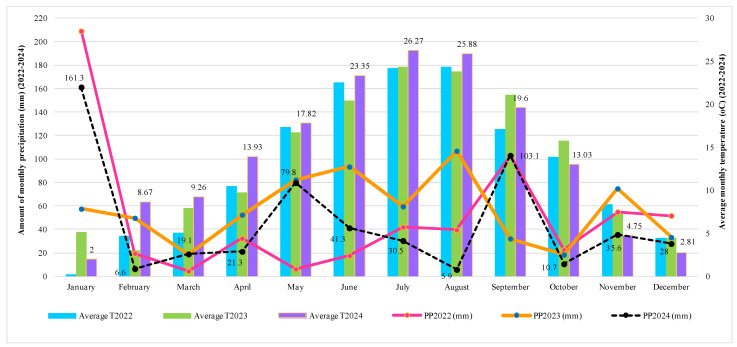
Temperature and precipitation in Timisoara area, Romania (2022–2024). (Average T2022, Average T2023, and Average T2024 → These refer to the mean air temperatures recorded during the growing seasons of 2022, 2023, and 2024, respectively. PP2022 (mm), PP2023 (mm), and PP2024 (mm) → These represent the total precipitation (rainfall) amounts, expressed in millimeters (mm), for the same years (2022–2024).

**Figure 2 plants-14-02807-f002:**
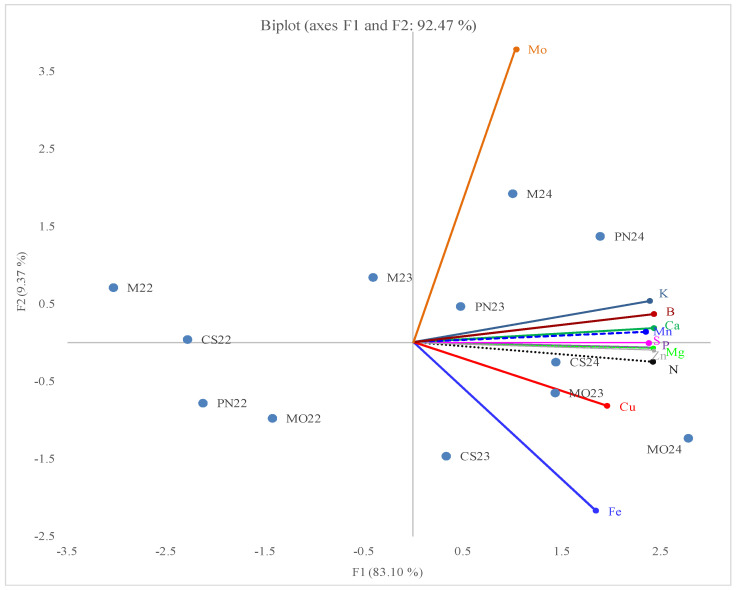
Principal component analysis (PCA) biplot of sap mineral composition across cultivars and years. (Blue points represent cultivar–year combinations (CS = ‘Cabernet Sauvignon’, PN = ‘Pinot Noir’, M = ‘Merlot’, MO = ‘Muscat Ottonel’; followed by the year: 22 = 2022, 23 = 2023, 24 = 2024). Colored vectors indicate mineral compound loadings: molybdenum (Mo, orange), potassium (K, navy blue), boron (B, burgundy), calcium (Ca, green), manganese (Mn, blue dots), sulphur (S, electric pink), phosphorus (P, purple), magnesium (Mg, electric green dots), zinc (Zn, grey), nitrogen (N, black), copper (Cu, red), and iron (Fe, blue). The direction and length of each vector represent the contribution and correlation of that mineral to the first two principal components. Cultivar–year points positioned in the direction of a given vector are more strongly associated with higher concentrations of that mineral.

**Figure 3 plants-14-02807-f003:**
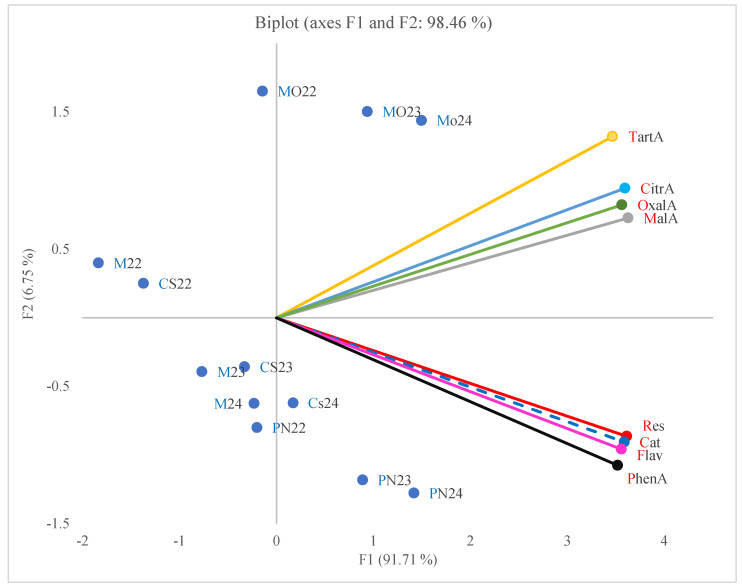
Principal component analysis (PCA) biplot of sap organic acid and polyphenolic composition across cultivars and years (Blue points represent cultivar–year combinations (CS = ‘Cabernet Sauvignon’, PN = ‘Pinot Noir’, M = ‘Merlot’, MO = ‘Muscat Ottonel’; followed by the year: 22 = 2022, 23 = 2023, 24 = 2024). Colored vectors indicate compound loadings: tartaric acid (TartA, yellow), citric acid (CitrA, light blue), oxalic acid (OxalA, green), malic acid (MalA, grey), resveratrol (Res, red), catechins (Cat, blue dots), flavonoids (Flav, purple), and phenolic acids (PhenA, black). The direction and length of each vector represent the contribution and correlation of that compound to the first two principal components. Cultivar–year points located in the direction of a given vector are more strongly associated with higher concentrations of that compound.

**Table 1 plants-14-02807-t001:** Onset of sap bleeding for each cultivar (2022–2024).

Cultivar	*n*	2022 Onset (±SD)	2023 Onset (±SD)	2024 Onset (±SD)
‘Cabernet Sauvignon’	30	March 20 ± 1.3 days	March 6 ± 1.2 days	March 4 ± 1.1 days
‘Merlot’	30	March 19 ± 1.5 days	March 5 ± 1.0 days	March 3 ± 0.9 days
‘Muscat Ottonel’	30	March 15 ± 1.4 days	March 3 ± 1.1 days	February 28 ± 1.1 days
‘Pinot Noir’	30	March 14 ± 1.2 days	March 2 ± 0.8 days	February 26 ± 0.9 days

SD = standard deviation; *n* = sample number.

**Table 2 plants-14-02807-t002:** Duration of sap bleeding for each cultivar (2022–2024).

Cultivar	*n*	2022 Duration (±SD)	2023 Duration (±SD)	2024 Duration (±SD)
‘Cabernet Sauvignon’	30	11 ± 2.0 days	16 ± 2.1 days	17 ± 1.6 days
‘Merlot’	30	10 ± 1.5 days	14 ± 2.0 days	15 ± 1.7 days
‘Muscat Ottonel’	30	13 ± 1.9 days	18 ± 2.0 days	20 ± 2.1 days
‘Pinot Noir’	30	14 ± 2.1 days	20 ± 1.8 days	21 ± 1.6 days

SD = standard deviation; *n* = sample number.

**Table 3 plants-14-02807-t003:** Bleeding intensity and several components (total soluble solids, pH, and electrical conductivity) of sap across cultivars and seasons (2022–2024).

Trait	Cultivar	*n*	2022 (±SD)	2023 (±SD)	2024 (±SD)	ANOVA *p*-Value
Bleeding intensity (mL/vine/day)	‘Cabernet Sauvignon’	30	4.9 ± 0.8 ^c^	6.1 ± 0.7 ^b^	6.8 ± 0.8 ^a^	0.027
	‘Merlot’	30	4.3 ± 0.6 ^c^	5.7 ± 0.6 ^b^	6.4 ± 0.7 ^a^	0.015
	‘Muscat Ottonel’	30	7.1 ± 0.7 ^c^	8.5 ± 1.0 ^b^	9.1 ± 1.1 ^a^	0.031
	‘Pinot Noir’	30	8.2 ± 0.9 ^c^	9.4 ± 1.1 ^b^	10.8 ± 1.2 ^a^	0.012
Total Soluble Solids (TSS, °Brix)	‘Cabernet Sauvignon’	30	2.1 ± 0.13 ^b^	2.3 ± 0.14 ^a^	2.3 ± 0.13 ^a^	0.032
	‘Merlot’	30	1.3 ± 0.12 ^c^	1.9 ± 0.13 ^b^	2.2 ± 0.14 ^a^	0.018
	‘Muscat Ottonel’	30	3.3 ± 0.14 ^c^	3.8 ± 0.13 ^ab^	3.8 ± 0.12 ^ab^	0.009
	‘Pinot Noir’	30	2.3 ± 0.12 ^a^	2.4 ± 0.13 ^a^	2.4 ± 0.14 ^a^	0.051 ^(ns)^
Sap pH	‘Cabernet Sauvignon’	30	6.3 ± 0.11 ^a^	6.6 ± 0.11 ^a^	6.6 ± 0.20 ^a^	0.513 ^(ns)^
	‘Merlot’	30	6.1 ± 0.21 ^a^	6.3 ± 0.20 ^a^	6.4 ± 0.12 ^a^	0.436 ^(ns)^
	‘Muscat Ottonel’	30	6.5 ± 0.11 ^a^	6.8 ± 0.22 ^a^	6.9 ± 0.21 ^a^	0.388 ^(ns)^
	‘Pinot Noir’	30	6.2 ± 0.12 ^a^	6.5 ± 0.21 ^a^	6.5 ± 0.20 ^a^	0.482 ^(ns)^
Electrical Conductivity (EC, mS/cm) *	‘Cabernet Sauvignon’	30	0.56 ± 0.06 ^b^	0.63 ± 0.05 ^a^	0.61 ± 0.04 ^a^	0.022
	‘Merlot’	30	0.52 ± 0.05 ^c^	0.60 ± 0.04 ^ab^	0.58 ± 0.05 ^b^	0.028
	‘Muscat Ottonel’	30	0.43 ± 0.04 ^b^	0.46 ± 0.05 ^a^	0.44 ± 0.03 ^b^	0.039
	‘Pinot Noir’	30	0.49 ± 0.05 ^a^	0.52 ± 0.06 ^a^	0.51 ± 0.04 ^a^	0.061 ^(ns)^

* mS = millisiemens (1 mS = 0.001 siemens); SD = standard deviation; *n* = sample number; means followed by different letters within a row are significantly different at *p* < 0.05 according to Tukey’s HSD. “^ns^” = not significant.

**Table 4 plants-14-02807-t004:** Macronutrient composition.

Cultivar	Year	*n*	N ± SD	P ± SD	K ± SD	Ca ± SD	Mg ± SD	S ± SD
‘Cabernet Sauvignon’	2022	30	49.1 ± 3.2 ^a^	4.4 ± 0.3 ^a^	118.2 ± 6.0 ^a^	34.6 ± 2.2 ^a^	8.8 ± 0.5 ^a^	5.1 ± 0.4 ^a^
	2023	30	56.7 ± 3.4 ^b^	5.2 ± 0.6 ^b^	134.8 ± 6.7 ^b^	38.1 ± 2.6 ^b^	9.6 ± 0.7 ^b^	5.4 ± 0.5 ^b^
	2024	30	58.5 ± 4.0 ^b^	5.5 ± 0.4 ^b^	139.2 ± 7.1 ^b^	39.4 ± 2.5 ^b^	10.2 ± 0.6 ^c^	5.6 ± 0.7 ^b^
ANOVA *p*-value			0.019	0.023	0.016	0.025	0.031	0.033
‘Merlot’	2022	30	47.2 ± 2.8 ^a^	4.4 ± 0.5 ^a^	120.1 ± 5.7 ^a^	33.8 ± 2.0 ^a^	8.6 ± 0.4 ^a^	4.8 ± 0.3 ^a^
	2023	30	54.5 ± 3.3 ^b^	4.8 ± 0.4 ^ab^	135.6 ± 7.2 ^b^	37.3 ± 2.3 ^b^	9.6 ± 0.5 ^b^	5.3 ± 0.4 ^b^
	2024	30	57.7 ± 3.6 ^c^	5.4 ± 0.5 ^b^	140.7 ± 6.6 ^b^	38.5 ± 2.4 ^b^	9.8 ± 0.7 ^b^	5.7 ± 0.4 ^b^
ANOVA *p*-value			0.015	0.018	0.013	0.021	0.027	0.030
‘Muscat Ottonel’	2022	30	53.6 ± 3.2 ^a^	4.8 ± 0.6 ^a^	124.2 ± 6.1 ^a^	35.6 ± 2.5 ^a^	9.2 ± 0.7 ^a^	5.1 ± 0.5 ^a^
	2023	30	60.2 ± 3.5 ^b^	5.4 ± 0.5 ^b^	141.7 ± 6.3 ^b^	39.2 ± 2.7 ^b^	10.2 ± 0.8 ^b^	5.7 ± 0.4 ^b^
	2024	30	62.8 ± 4.3 ^b^	5.8 ± 0.7 ^b^	145.1 ± 7.0 ^b^	40.2 ± 2.5 ^b^	10.5 ± 0.6 ^b^	6.1 ± 0.5 ^c^
ANOVA *p*-value			0.011	0.016	0.015	0.017	0.026	0.028
‘Pinot Noir’	2022	30	51.7 ± 2.6 ^a^	4.6 ± 0.3 ^a^	122.6 ± 6.1 ^a^	34.2 ± 2.2 ^a^	8.7 ± 0.5 ^a^	5.2 ± 0.4 ^a^
	2023	30	58.4 ± 3.1 ^b^	5.2 ± 0.5 ^b^	138.4 ± 7.2 ^b^	38.6 ± 2.6 ^b^	9.8 ± 0.7 ^b^	5.5 ± 0.6 ^b^
	2024	30	61.2 ± 3.8 ^b^	5.6 ± 0.4 ^b^	142.2 ± 6.7 ^b^	39.7 ± 2.5 ^b^	10.1 ± 0.8 ^b^	5.8 ± 0.5 ^b^
ANOVA *p*-value			0.017	0.022	0.014	0.017	0.024	0.031

Superscript letters denote statistical groupings for each nutrient within a cultivar (Tukey HSD); Different letters (a, b, c) = significant difference at *p* < 0.05; The differences align with your *p*-values, indicating clear year-to-year changes in nutrient content; SD = standard deviation; *n* = sample number.

**Table 5 plants-14-02807-t005:** Micronutrient composition in bleeding sap.

Cultivar	Year	*n*	Fe ± SD	Mn ± SD	Zn ± SD	Cu ± SD	B ± SD	Mo ± SD
‘Cabernet Sauvignon’	2022	30	1.2 ± 0.2 ^b^	0.31 ± 0.02 ^c^	0.51 ± 0.06 ^c^	0.18 ± 0.02 ^b^	0.80 ± 0.07 ^c^	0.04 ± 0.02 ^a^
	2023	30	1.4 ± 0.2 ^a^	0.35 ± 0.03 ^b^	0.61 ± 0.05 ^b^	0.20 ± 0.01 ^a^	0.92 ± 0.06 ^b^	0.03 ± 0.01 ^a^
	2024	30	1.3 ± 0.1 ^ab^	0.38 ± 0.04 ^a^	0.64 ± 0.04 ^a^	0.21 ± 0.02 ^a^	1.01 ± 0.06 ^a^	0.04 ± 0.02 ^a^
ANOVA *p*-value			0.021	0.027	0.026	0.035	0.018	0.030
‘Merlot’	2022	30	1.1 ± 0.1 ^b^	0.29 ± 0.02 ^c^	0.51 ± 0.03 ^c^	0.15 ± 0.01 ^b^	0.79 ± 0.06 ^c^	0.04 ± 0.02 ^b^
	2023	30	1.3 ± 0.2 ^a^	0.33 ± 0.04 ^b^	0.57 ± 0.05 ^b^	0.17 ± 0.02 ^ab^	0.92 ± 0.05 ^b^	0.05 ± 0.01 ^ab^
	2024	30	1.2 ± 0.1 ^ab^	0.38 ± 0.03 ^a^	0.62 ± 0.06 ^a^	0.19 ± 0.02 ^a^	0.97 ± 0.05 ^a^	0.06 ± 0.02 ^a^
ANOVA *p*-value			0.021	0.025	0.023	0.031	0.019	0.028
‘Muscat Ottonel’	2022	30	1.3 ± 0.2 ^b^	0.32 ± 0.04 ^c^	0.54 ± 0.06 ^c^	0.17 ± 0.02 ^b^	0.86 ± 0.06 ^c^	0.03 ± 0.02 ^a^
	2023	30	1.4 ± 0.2 ^b^	0.36 ± 0.02 ^b^	0.62 ± 0.05 ^b^	0.21 ± 0.01 ^a^	0.98 ± 0.05 ^b^	0.04 ± 0.01 ^a^
	2024	30	1.6 ± 0.1 ^a^	0.41 ± 0.03 ^a^	0.68 ± 0.06 ^a^	0.20 ± 0.02 ^a^	1.03 ± 0.04 ^a^	0.04 ± 0.02 ^a^
ANOVA *p*-value			0.016	0.024	0.022	0.027	0.021	0.031
‘Pinot Noir’	2022	30	1.2 ± 0.2 ^b^	0.28 ± 0.01 ^c^	0.54 ± 0.04 ^c^	0.18 ± 0.02 ^ab^	0.81 ± 0.05 ^c^	0.03 ± 0.02 ^b^
	2023	30	1.4 ± 0.2 ^a^	0.34 ± 0.02 ^b^	0.60 ± 0.05 ^b^	0.17 ± 0.01 ^b^	0.93 ± 0.06 ^b^	0.05 ± 0.02 ^ab^
	2024	30	1.3 ± 0.1 ^ab^	0.37 ± 0.05 ^a^	0.65 ± 0.06 ^a^	0.21 ± 0.02 ^a^	1.02 ± 0.06 ^a^	0.06 ± 0.01 ^a^
ANOVA *p*-value			0.018	0.023	0.021	0.028	0.021	0.026

Different lowercase letters within each cultivar and column indicate significant differences between years (*p* < 0.05, Tukey HSD); SD = standard deviation; *n* = sample number.

**Table 6 plants-14-02807-t006:** Total protein content and activities of peroxidase, polyphenol oxidase, and β-glucosidase in bleeding sap (2022–2024).

Trait	Cultivar	*n*	2022 (±SD)	2023 (±SD)	2024 (±SD)	ANOVA *p*-Value
Total protein content (mg/mL)	‘Cabernet Sauvignon’	30	0.62 ± 0.07 ^c^	0.75 ± 0.06 ^b^	0.79 ± 0.05 ^a^	0.018
	‘Merlot’	30	0.59 ± 0.06 ^c^	0.72 ± 0.05 ^b^	0.77 ± 0.06 ^a^	0.015
	‘Muscat Ottonel’	30	0.84 ± 0.08 ^c^	0.93 ± 0.07 ^b^	0.95 ± 0.06 ^a^	0.032
	‘Pinot Noir’	30	0.89 ± 0.09 ^c^	0.97 ± 0.06 ^b^	1.05 ± 0.07 ^a^	0.022
Peroxidase activity (U/mL) *	‘Cabernet Sauvignon’	30	5.4 ± 0.5 ^c^	6.8 ± 0.6 ^b^	7.1 ± 0.6 ^a^	0.021
	‘Merlot’	30	4.9 ± 0.4 ^c^	6.1 ± 0.5 ^b^	6.6 ± 0.6 ^a^	0.016
	‘Muscat Ottonel’	30	6.2 ± 0.5 ^c^	7.0 ± 0.6 ^b^	7.4 ± 0.7 ^a^	0.039
	‘Pinot Noir’	30	6.9 ± 0.6 ^c^	7.6 ± 0.6 ^b^	8.1 ± 0.8 ^a^	0.025
Polyphenol oxidase activity (U/mL) **	‘Cabernet Sauvignon’	30	1.8 ± 0.2 ^c^	2.2 ± 0.2 ^b^	2.4 ± 0.2 ^a^	0.030
	‘Merlot’	30	1.6 ± 0.2 ^c^	2.1 ± 0.3 ^b^	2.2 ± 0.2 ^a^	0.028
	‘Muscat Ottonel’	30	2.1 ± 0.3 ^c^	2.5 ± 0.2 ^b^	2.7 ± 0.2 ^a^	0.041
	‘Pinot Noir’	30	2.3 ± 0.2 ^c^	2.7 ± 0.3 ^b^	3.0 ± 0.3 ^a^	0.020
β-glucosidase activity (U/mL) *	‘Cabernet Sauvignon’	30	0.94 ± 0.08 ^c^	1.18 ± 0.09 ^b^	1.25 ± 0.08 ^a^	0.019
	‘Merlot’	30	0.87 ± 0.07 ^c^	1.10 ± 0.08 ^b^	1.15 ± 0.09 ^a^	0.015
	‘Muscat Ottonel’	30	1.05 ± 0.09 ^c^	1.24 ± 0.10 ^b^	1.30 ± 0.09 ^a^	0.026
	‘Pinot Noir’	30	1.10 ± 0.08 ^c^	1.31 ± 0.11 ^b^	1.38 ± 0.10 ^a^	0.018

* U = Enzyme Unit (the amount of enzyme that catalyzes the conversion of one micromole (µmol) of substrate per minute under assay conditions); ** U = enzyme unit; identical letters indicate no significant difference; SD = standard deviation; *n* = sample number.

**Table 7 plants-14-02807-t007:** Organic acid content in xylem sap with ANOVA *p*-values (2022–2024).

Cultivar	Year	*n*	Malic Acid ± SD	Tartaric Acid ± SD	Citric Acid ± SD	Oxalic Acid ± SD
‘Cabernet Sauvignon’	2022	30	72.3 ± 4.0 ^c^	46.2 ± 2.6 ^c^	18.4 ± 1.3 ^c^	9.2 ± 0.5 ^c^
	2023	30	79.1 ± 3.7 ^b^	49.6 ± 2.4 ^b^	21.0 ± 1.5 ^b^	9.7 ± 0.4 ^b^
	2024	30	81.5 ± 4.1 ^a^	50.8 ± 2.7 ^a^	22.2 ± 1.6 ^a^	10.2 ± 0.6 ^a^
ANOVA *p*-value			0.017	0.041	0.021	0.035
‘Merlot’	2022	30	69.7 ± 3.8 ^c^	44.4 ± 2.2 ^c^	17.5 ± 1.2 ^c^	8.8 ± 0.3 ^c^
	2023	30	75.5 ± 3.6 ^b^	47.7 ± 2.5 ^b^	19.7 ± 1.3 ^b^	9.5 ± 0.5 ^b^
	2024	30	78.0 ± 4.0 ^a^	49.1 ± 2.8 ^a^	20.6 ± 1.5 ^a^	10.2 ± 0.4 ^a^
ANOVA *p*-value			0.021	0.037	0.024	0.030
‘Muscat Ottonel’	2022	30	84.6 ± 4.1 ^c^	52.5 ± 3.1 ^c^	23.2 ± 1.3 ^c^	10.1 ± 0.5 ^c^
	2023	30	91.1 ± 3.9 ^b^	56.0 ± 3.2 ^b^	25.6 ± 1.5 ^b^	11.3 ± 0.7 ^b^
	2024	30	94.4 ± 4.5 ^a^	57.3 ± 3.0 ^a^	27.1 ± 1.6 ^a^	12.0 ± 0.8 ^a^
ANOVA *p*-value			0.013	0.027	0.017	0.021
‘Pinot Noir’	2022	30	81.2 ± 4.0 ^c^	48.8 ± 2.7 ^c^	21.0 ± 1.5 ^c^	9.7 ± 0.5 ^c^
	2023	30	87.5 ± 4.2 ^b^	52.2 ± 3.1 ^b^	23.8 ± 1.3 ^b^	10.6 ± 0.4 ^b^
	2024	30	91.0 ± 4.4 ^a^	53.4 ± 2.8 ^a^	25.2 ± 1.6 ^a^	11.2 ± 0.3 ^a^
ANOVA *p*-value			0.018	0.031	0.020	0.024

Different lowercase letters within each cultivar and column indicate significant differences between years (*p* < 0.05, Tukey HSD). SD = standard deviation; *n* = sample number.

**Table 8 plants-14-02807-t008:** Total phenolic compound content in bleeding sap (2022–2024).

Cultivar	Year	*n*	Resveratrol ± SD	Catechins ± SD	Flavonoids ± SD	Phenolic Acids ± SD
‘Cabernet Sauvignon’	2022	30	1.84 ± 0.11 ^c^	3.41 ± 0.17 ^c^	5.16 ± 0.25 ^c^	2.63 ± 0.13 ^c^
	2023	30	2.13 ± 0.14 ^b^	3.67 ± 0.20 ^b^	5.45 ± 0.27 ^b^	2.91 ± 0.15 ^b^
	2024	30	2.24 ± 0.16 ^a^	3.88 ± 0.22 ^a^	5.57 ± 0.24 ^a^	3.00 ± 0.18 ^a^
ANOVA *p*-values			0.018	0.021	0.026	0.031
‘Merlot’	2022	30	1.71 ± 0.10 ^c^	3.34 ± 0.16 ^c^	5.02 ± 0.23 ^c^	2.50 ± 0.12 ^c^
	2023	30	2.00 ± 0.13 ^b^	3.58 ± 0.19 ^b^	5.36 ± 0.26 ^b^	2.82 ± 0.16 ^b^
	2024	30	2.11 ± 0.15 ^a^	3.80 ± 0.21 ^a^	5.47 ± 0.22 ^a^	2.93 ± 0.17 ^a^
ANOVA *p*-values			0.015	0.020	0.023	0.025
‘Muscat Ottonel’	2022	30	2.04 ± 0.14 ^c^	3.67 ± 0.20 ^c^	5.43 ± 0.24 ^c^	2.72 ± 0.15 ^c^
	2023	30	2.27 ± 0.16 ^b^	3.88 ± 0.21 ^b^	5.70 ± 0.28 ^b^	3.03 ± 0.18 ^b^
	2024	30	2.39 ± 0.17 ^a^	4.03 ± 0.24 ^a^	5.82 ± 0.23 ^a^	3.16 ± 0.20 ^a^
ANOVA *p*-values			0.018	0.020	0.024	0.028
‘Pinot Noir’	2022	30	2.14 ± 0.13 ^c^	3.78 ± 0.20 ^c^	5.63 ± 0.26 ^c^	2.86 ± 0.14 ^c^
	2023	30	2.40 ± 0.15 ^b^	4.04 ± 0.23 ^b^	5.92 ± 0.29 ^b^	3.15 ± 0.17 ^b^
	2024	30	2.51 ± 0.16 ^a^	4.17 ± 0.24 ^a^	6.05 ± 0.25 ^a^	3.28 ± 0.20 ^a^
ANOVA *p*-values			0.017	0.021	0.027	0.031

Different lowercase letters within each cultivar and column indicate significant differences between years (*p* < 0.05, Tukey HSD).

**Table 9 plants-14-02807-t009:** Parameters for chemical analyses of grapevine sap.

Analysis	Column and Conditions	Mobile Phase/Gradient	Detection	Standards and Calibration	LOD/LOQ	QC/Blanks
Ion Chromatography (IC)—anions (NO_3_^−^, PO_4_^3−^, SO_4_^2−^, Cl^−^)	Dionex IonPac AS11-HC (4 × 250 mm) + AG11-HC guard; 30 °C; 1.0 mL·min^−1^; injection 25 µL	KOH gradient: 10 mM (0–5 min) → 30 mM (5–15 min) → 60 mM (15–25 min) → 10 mM (25–30 min, re-eq.)	Suppressed conductivity (AERS 500, 78 mA; Thermo Fisher Scientific Inc. 168 Third Avenue Waltham, MA USA)	Certified standards: 0.05–50 mg L^−1^, 7 points, r^2^ ≥ 0.999	0.01–0.05/0.03–0.15 mg L^−1^	Method blank and CCV every 10 samples; matrix spikes (80–120% recovery); duplicate injections (RSD ≤ 5%)
Ion Chromatography (IC)—cations (K^+^, Ca^2+^, Mg^2+^, NH_4_^+^, Na^+^)	Dionex IonPac CS12A (4 × 250 mm) + CG12A guard; 30 °C; 1.0 mL·min^−1^; injection 25 µL	20 mM MSA, isocratic	Suppressed conductivity (CERS 500, 59 mA; Thermo Fisher Scientific Inc. 168 Third Avenue Waltham, MA USA)	Same as above; 0.02–50 mg L^−1^, 7 points	0.02–0.05/0.06–0.15 mg L^−1^	As above
HPLC (organic acids: malic, tartaric, citric, oxalic)	Bio-Rad Aminex HPX-87H (300 × 7.8 mm) + guard; 65 °C; 0.5 mL·min^−1^; injection 20 µL	5 mM H_2_SO_4_, isocratic	UV 210 nm	Authentic standards: 0.05–20 mg L^−1^, r^2^ ≥ 0.999	0.01–0.03/0.03–0.10 mg L^−1^	Blanks; CCV every 10 injections; duplicate injections (RSD ≤ 5%); system suitability N ≥ 5000
HPLC-DAD (phenolics: phenolic acids, flavonoids, catechins, resveratrol)	C18 column (250 × 4.6 mm, 5 µm) + guard; 30 °C; 1.0 mL·min^−1^; injection 10 µL	A = H_2_O + 0.1% formic acid; B = ACN + 0.1% formic acid; gradient: 5% B (0 min) → 95% B (37 min), re-eq. 10 min	DAD: 280 nm (catechins), 306–310 nm (resveratrol), 320 nm (hydroxycinnamates), 360 nm (flavonols); spectra 200–400 nm	Authentic standards (gallic, caffeic, ferulic acids, catechin, quercetin, resveratrol); 0.05–25 mg L^−1^, r^2^ ≥ 0.998	0.005–0.02/0.02–0.06 mg L^−1^	Mobile phase blanks; CCV every 10; matrix spikes (80–120%); duplicate injections; retention-time window ±2%; spectral match ≥0.98
Bradford assay (protein)	—	—	UV-Vis at 595 nm (Shimadzu UV-1800)	BSA calibration curve 0–1.0 mg·mL^−1^, r^2^ ≥ 0.999		

IC = Ion Chromatography; HPLC = High-Performance Liquid Chromatography; HPLC-DAD = High-Performance Liquid Chromatography with Diode-Array Detection; UV-Vis = Ultraviolet–Visible Spectrophotometry; UV = Ultraviolet; DAD = Diode-Array Detector; KOH = Potassium hydroxide; MSA = Methanesulfonic acid; H_2_SO_4_ = Sulfuric acid; ACN = Acetonitrile; C18 = Octadecylsilane reversed-phase column; AERS/CERS 500 = Anion/Cation Electrolytically Regenerated Suppressor; BSA = Bovine Serum Albumin; r^2^ = coefficient of determination; LOD = Limit of Detection; LOQ = Limit of Quantification; QC = Quality Control; CCV = Continuing Calibration Verification; RSD = Relative Standard Deviation; N = theoretical plate number (column efficiency); NO_3_^−^ = Nitrate; PO_4_^3−^ = Phosphate; SO_4_^2−^ = Sulfate; Cl^−^ = Chloride; K^+^ = Potassium; Ca^2+^ = Calcium; Mg^2+^ = Magnesium; NH_4_^+^ = Ammonium; Na^+^ = Sodium.

## Data Availability

No new data were created or analyzed in this study. Data sharing is not applicable to this article.

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
