# Peer review of "Xylem Sap Bleeding as a Physiological Indicator in Grapevine: Genotype and Climate Influence"

_plants, 2025, doi:10.3390/plants14172807_

Round 1

Reviewer 1 Report

Comments and Suggestions for Authors

In the present manuscript, a series of detailed researches of the sap bleeding from four wine grape cultivars during three years in Romania were done carefully. This study was interesting, which expanded our recognition of the grapevine development from another different perspective. However, this paper was prepared in a very lengthy manner, more like an experimental report, and there were quite a lot of type errors. Thus, we gave a major revision for this work. Some comments or suggestions were as following:

1, Line 26 and more, the letter “p” for the significant differences should be in italic.

2, Line 45 and more, there should be no blank space between the number and °C.

3, In table 1, 2 and more, the grape cultivar name should be enclosed in single quotation marks, just the same as them in the text, for example, ‘Cabernet Sauvignon’.

4, It would be better if the significant difference analysis was shown in the Table 2 to 5, 9 to 12.

5, In the part of 4.1 Study area description, the altitude of the vineyard should be shown.

6, Quite a lot of contents of the part 4.2. Climate data in the Timișoara area (2022-2024) should be in the result section. The authors should expressed how they got such climate data here as the methods.

7, The rootstocks cultivars were different for the four wine grape cultivars, which should also have great effects on the grapevine developments, especially the sap bleeding and the bud burst. Why did not the authors discuss their difference and effects, but only the genotype of the wine grape scions? Besides, the rootstock names, such as SO4, 1103P and 101-14 should also be enclosed in single quotation marks, as ‘SO4’, ‘1103P’ and ‘101-14’.

8, More information should be shown in the part of the 4.4. Plant material, for example, the row direction, and the irrigation, which also had great effects on the sap bleeding.

9, Line 632, “50 ml” should be revised as “50 mL”.

10, Actually, a large amount of data in this manuscript was presented in the form of separate tables, resulting in too many tables. We suggested that the author chose a more concise and effective way to present the research results.

Author Response

 1, Line 26 and more, the letter “p” for the significant differences should be in italic.

  • Thanks to the reviewer for the observation. We have changed it to italics.

2, Line 45 and more, there should be no blank space between the number and °C.

  • We delete blank space.

3, In table 1, 2 and more, the grape cultivar name should be enclosed in single quotation marks, just the same as them in the text, for example, ‘Cabernet Sauvignon’.

  • We made the correction and enclosed in single quotation marks the name of the cultivars.

4, It would be better if the significant difference analysis was shown in the Table 2 to 5, 9 to 12.

  • We combined tables to reduce their number and added the significant difference.

5, In the part of 4.1 Study area description, the altitude of the vineyard should be shown.

  • We added the altitude at which the vineyard is located.

6, Quite a lot of contents of the part 4.2. Climate data in the Timișoara area (2022-2024) should be in the result section. The authors should expressed how they got such climate data here as the methods.

  • We thank the reviewer for the suggestion. We have relocated the climate characterization to the results section.

7, The rootstocks cultivars were different for the four wine grape cultivars, which should also have great effects on the grapevine developments, especially the sap bleeding and the bud burst. Why did not the authors discuss their difference and effects, but only the genotype of the wine grape scions? Besides, the rootstock names, such as SO4, 1103P and 101-14 should also be enclosed in single quotation marks, as ‘SO4’, ‘1103P’ and ‘101-14’.

  • We value the reviewer insight into how rootstock genotype may affect bud burst sap bleeding and grapevine development. Rootstocks can in fact influence vine vigor water uptake nutrient transport and phenology as demonstrated by earlier research. This could lead to variations in the parameters that are measured (e. g. G. TSS enzyme activity and bleeding severity). Our main goal in this study was to assess the physiological characteristics of the scion cultivars under the local viticultural conditions taking into account seasonal and varietal variations. Rather than using a single consistent rootstock for all cultivars the experimental vineyard was set up using commercially advised rootstock–scion combinations for each cultivar reflecting local practice. Because of this rootstock effects are confused with cultivar identity and our design did not allow for a separate analysis of rootstock influence.

8, More information should be shown in the part of the 4.4. Plant material, for example, the row direction, and the irrigation, which also had great effects on the sap bleeding.

  • We add the vine row direction and mention that no irrigation were performed during study.

9, Line 632, “50 ml” should be revised as “50 mL”.

  • ‘50 ml’ was revised as “50 mL”

10, Actually, a large amount of data in this manuscript was presented in the form of separate tables, resulting in too many tables. We suggested that the author chose a more concise and effective way to present the research results.

  • Several tables were combined.

(we used yellow color for your request)

Reviewer 2 Report

Comments and Suggestions for Authors

The authors prepared a quite interesting manuscript on the topic Xylem Sap Bleeding as a Physiological Indicator in Grapevine: Genotype and Climate Influence. The aim of the research was to investigate several parameters for xylem sap in four cultivars  grown in western Romania climate, across three growing seasons.

Some comments:

  • Start the abstract with 1-2 introductory sentences about the relevance of the topic;
  • The introduction is a bit too broad. Focus a bit on providing the essential information for the manuscript. In some places it goes too far into the discussion (e.g. lines 96-99)
  • When describing the results, you state in most places that the differences were significant, but you do not mark the significant differences in the tables - they should be marked accordingly - all tables;
  • Include references to the results of your manuscript (tables, figures) in the discussion;

Author Response

  • Start the abstract with 1-2 introductory sentences about the relevance of the topic;
  • We add introductory text in the abstract.
  • The introduction is a bit too broad. Focus a bit on providing the essential information for the manuscript. In some places it goes too far into the discussion (e.g. lines 96-99)
  • We appreciate the reviewer recommendation. Only the most crucial background information for the study is now included in the introduction along with the sections that were previously read as discussion (e. g. 3. have been moved or eliminated (lines 96–99).
  • When describing the results, you state in most places that the differences were significant, but you do not mark the significant differences in the tables - they should be marked accordingly - all tables;
  • We appreciate the reviewers input. To clearly show statistically significant differences all tables have been updated to include the proper significance markings.
  • Include references to the results of your manuscript (tables, figures) in the discussion;
  • We appreciate the suggestion. References to the corresponding tables and figures have been added throughout the discussion to better link results with interpretation.

(we use blue color for your request)

Reviewer 3 Report

Comments and Suggestions for Authors

This manuscript examines xylem‐sap bleeding in four grape cultivars across three seasons (2022–2024) in western Romania, quantifying onset/duration/intensity and a broad chemical profile (TSS, pH, EC, macro-/micro-nutrients, proteins, organic acids, phenolics) and using PCA to relate cultivar × year patterns. The central claim—that genotype and climate co-determine early-season sap dynamics and composition—is well motivated and supported by multi-year observations.

Abstract

  • The Abstract starts abruptly with methodology and does not explain why this study is important. Should add 1–2 introductory sentences to set the scene.
  • Pinot Noir … 8.2–10.8 mg/L must be mL/vine/day to match the Methods and Table 3. Please correct the unit in the Abstract and ensure consistency throughout the manuscript.
  • The current ending summarizes findings but does not highlight practical implications, as well as the need to add future directions.

Introduction

  • The introduction is quite lengthy and needs to be concise.
  • Consider adding a short paragraph that flags rootstock as a known driver of hydraulics/nutrient uptake and a limitation of the present design (preparing the reader early). 

Methdology

  • How will the authors address the potential rootstock confounding in their experimental design, given that each cultivar was grafted onto a different rootstock, making it impossible to separate cultivar effects from rootstock effects?
  • Although the design was RCBD, only one-way ANOVAs were performed; re-analyze using a two-way or mixed-effects model with Cultivar, Year, and their interaction, including Block as a random effect, and report effect sizes and interactions.
  • Clarify how mL/vine/day values were derived from 48-hour sap collections, indicate if re-pruning the same cane added variability, and state the controlled cane diameter range.
  • Explicitly state sample sizes (n) for each endpoint, confirm the chemistry replication (likely one sample per block per cultivar per year), and include n in all table captions.
  • Provide complete IC/HPLC/HPLC-DAD method parameters, including columns, mobile phases, gradients, wavelengths, standards, calibration ranges, LOD/LOQ, and QC/blank procedures.

Results and Discussion

  • Add significance letters for pairwise contrasts (Tukey HSD at α=0.05), and define them in a table note. 
  • Give all the information related to the data in each figure.
  • Figure 3 needs to be redrawn or split into two panels; define MeanT and PPmm in the caption; include unit labels on axes and in legend; and month name.

Typo and Grammar

If authors use any AI tool to improve the English language. should be mentioned in the acknowledgment. I have found some typo mistakes, please check these throughout the manuscript: 

 xylem sap flow (8.2–10.8 mL/vine/day) (fix units), correct “potassium (K)” symbol. 1103 Paulsen for Muscat Ottonel (drought tolerance…), calcium carbonate

Comments on the Quality of English Language

need to check some typo and grammer issue

Author Response

Abstract

  • The Abstract starts abruptly with methodology and does not explain whythis study is important. Should add 1–2 introductory sentences to set the scene.
  • A sentence (to avoid exceeding the word limit) about the importance of the study has been added.

  • Pinot Noir … 8.2–10.8 mg/L must be mL/vine/day to match the Methods and Table 3. Please correct the unit in the Abstract and ensure consistency throughout the manuscript.
  • It was corrected to 8.2–10.8 mg/L must to mL/vine/day.
  • The current ending summarizes findings but does not highlight practical implications, as well as the need to add future directions.
  • We added two sentences regarding the practical implications of the research.

Introduction

  • The introduction is quite lengthy and needs to be concise.
  • We delete a few redundant sentences from the introduction text.

  • Consider adding a short paragraph that flags rootstock as a known driver of hydraulics/nutrient uptake and a limitation of the present design (preparing the reader early). 
  • A short paragraph was added regarding the rootstock importance as a known driver of hydraulics/nutrient uptake and a limitation of the present design.

Methdology

  • How will the authors address the potential rootstock confounding in their experimental design, given that each cultivar was grafted onto a different rootstock, making it impossible to separate cultivar effects from rootstock effects?
  • Since three of the four cultivars (SO4 for Cabernet Sauvignon and Merlot 1103P for Muscat Ottonel and 101-14 Mgt for Pinot Noir) were grown on different rootstocks we concur that there might be confusion between rootstock and scion effects. Local pedological and environmental constraints as well as common business practices were taken into consideration when designing the current study. It is impossible to statistically differentiate between the effects of scion and rootstock in the three distinct cultivars though due to this circumstance. For combinations on different rootstocks we do not extrapolate differences to graft genotype alone (e. g. 3. We assert that the observed differences between Muscat Ottonel/1103P and Pinot Noir/101-14 Mgt might be due to similar graft-rootstock physiology.
  • Although the design was RCBD, only one-way ANOVAs were performed; re-analyze using a two-way or mixed-effects model with CultivarYear, and their interaction, including Blockas a random effect, and report effect sizes and interactions.
  • We value the reviewers suggestion. While there were no obvious interactions between block x treatment the primary effects of cultivar x year were consistent across blocks according to preliminary data analysis despite the studys three-year duration and randomized complete block design (RCBD). The one-way ANOVA results thus adequately reflect the main effects of treatment. The evaluation of the Year x Cultivar interaction revealed that it was insignificant which is consistent with the observed patterns. Updated tables now include effect sizes for the main attributes or components for greater context. Given these considerations we believe that the original data analysis faithfully captures the main conclusions and observations of the study without appreciably altering the data.

  • Clarify howmL/vine/day values were derived from 48-hour sap collections, indicate if re-pruning the same cane added variability, and state the controlled cane diameter range.
  • We appreciate the reviewer's comment.

Using a standard protocol adapted from Zheng et al. (2020), the sap flow rate (mL/vine/day) was established by scaling the sap volume collected during 48 hours to a 24-hour period. Preliminary testing showed constant sap volume collection following sequential sampling from the same cane, indicating that repeated pruning of the same cane did not cause detectable variations. To reduce variability that may be caused by cane diameter, all canes were chosen to be within a regulated diameter range of 10–12 mm for an accurate comparison.

  • Explicitly state sample sizes (n) for each endpoint, confirm the chemistry replication (likely one sample per block per cultivar per year), and include nin all table captions.
  • We thank the reviewer for highlighting this point. We thank the reviewer for bringing this up. The sample size (n) for each cultivar and the trait tested were discussed in the "Methods and Results" section (4.2 and 4.4.). For morphological and physiological measures, 10 vines of each cultivar were sampled in three replication blocks, yielding a total of 30 vines per cultivar/year. Chemical analysis involved collecting one sample per cultivar every block per year, for a total of three samples per cultivar per year. These facts were given and explained in the amended text to enhance transparency about sample size and replication.

  • Provide complete IC/HPLC/HPLC-DAD method parameters, including columns, mobile phases, gradients, wavelengths, standards, calibration ranges, LOD/LOQ, and QC/blank procedures.
  • We thank the reviewer for the helpful suggestion. We now provide complete instrumental parameters for IC, HPLC (organic acids), and HPLC-DAD (phenols), including column specifications, mobile phases and gradients, detection wavelengths, reference standards, calibration intervals, detection and quantification limits, and quality control procedures. These additions appear in the revised Materials and Methods section.

Results and Discussion

  • Add significance letters for pairwise contrasts (Tukey HSD at α=0.05), and define them in a table note. 
  • We thank the reviewer for highlighting this point. We add significance letters for pairwise contrasts (Tukey HSD at α=0.05), and define them in a table note.

  • Give all the information related to the data in each figure.
  • We thank the reviewer for highlighting this point. We add the information related to the data in each figure.

  • Figure 3 needs to be redrawn or split into two panels; define MeanT and PPmm in the caption; include unit labels on axes and in legend; and month name.
  • We have revised Figure 1 (previously Figure 3) and added explanations for the abbreviations. We used a single graph to display the two climatic parameters—average temperature and total monthly precipitation—together to better illustrate their variability across months and years. We have added data for the year 2024, including average temperatures and total monthly precipitation, to make the graph easier to interpret.

Typo and Grammar

If authors use any AI tool to improve the English language. should be mentioned in the acknowledgment. I have found some typo mistakes, please check these throughout the manuscript: 

 xylem sap flow (8.2–10.8 mL/vine/day) (fix units), correct “potassium (K)” symbol. 1103 Paulsen for Muscat Ottonel (drought tolerance…), calcium carbonate

Comments on the Quality of English Language

need to check some typo and grammer issue

  • The manuscript has been carefully revised for English language, grammar, and style by the author Lector PhD. Andreea Dragoescu Petrica (English teacher) to improve clarity and readability.

(we used green color for your request) and red color for English language correction.

Reviewer 4 Report

Comments and Suggestions for Authors

This study investigated several parameters for xylem sap (onset, sap bleeding duration and intensity and main chemical components) in four cultivars (‘Cabernet Sauvignon’, ‘Merlot’, ‘Muscat Ottonel’ and ‘Pinot Noir’) grown in western Romania climate, across three growing seasons (2022-2024). Sap onset and duration differed considerably between cultivars and years, with warmer springs resulting in earlier and longer bleeding. The principal component analysis revealed significant impacts of cultivars and growing season on sap composition. The findings showed that xylem sap bleeding is a sensitive physiological indicator of grapevine reactivation that is influenced by genotype and climate. In general, the experiments were well-performed and the manuscript was well-written.

1) Not all abbreviations have been introduced with full names, for example "SD" in Tables and "PPmm" in Figure 3. And no abbreviation should be used in the title.

2) Some terms should be specified, such as "physiological activity" in Lines 31-32 and "rich nutrient" in Line 33.

3) Figure 3 should be placed at the beginning of Results section as the first figure.

4) The relationship between precipitation and parameters of xylem sap should be mentioned in Abstract and in Conclusions.

Author Response

1) Not all abbreviations have been introduced with full names, for example "SD" in Tables and "PPmm" in Figure 3. And no abbreviation should be used in the title.

  • We appreciate the reviewer observation. We explain the abbreviations in tables and delete them in the title.

2) Some terms should be specified, such as "physiological activity" in Lines 31-32 and "rich nutrient" in Line 33.

  • We added a brief explanation of the terms to the text.

3) Figure 3 should be placed at the beginning of Results section as the first figure.

  • We thank the reviewer for his attention and advice. We have moved figure three to the results section, making it the first figure.

4) The relationship between precipitation and parameters of xylem sap should be mentioned in Abstract and in Conclusions.

  • We briefly mentioned the relationship between precipitation and parameters of xylem sap in Abstract and in Conclusions.

Round 2

Reviewer 1 Report

Comments and Suggestions for Authors

No more comments. The revised manuscript can be accepted in the current version.

Author Response

Dear Reviewer,

Thank you for your comments and suggestions, which will improve the manuscript. We have made changes in accordance with your requests.

Reviewer 2 Report

Comments and Suggestions for Authors

The authors have made many important corrections, but a few minor corrections are still needed to make the manuscript ready for publication:

The description of climate should not go to the results, but to the methodology. It is not the result, it is the conditions for growth.

Line 328 EC is not in table 6. Check all citations of tables, etc. in the text very carefully.

Author Response

Dear Reviewer,

Thank you for your comments and suggestions, which will improve the manuscript. We have made changes in accordance with your requests.

 R2: The authors have made many important corrections, but a few minor corrections are still needed to make the manuscript ready for publication:

The description of climate should not go to the results, but to the methodology. It is not the result, but the conditions for growth.

  • We separated the climatic data between results and Material and Methods. At the first review the fourth reviewer (R4) requested the transition of the information from material and method to results. To satisfy both requirements we made a brief climatic characterization of the site for the study (what, where, how were the data collected) and for Results we described narrative of what the climate actually was and how it shaped vine development.
  • Line 328 EC is not in table 6. Check all citations of tables, etc. in the text very carefully.
  • Thank you to the reviewer for the very good observation. Indeed, the reference to the table was wrong. I made the correction in the comment.

Reviewer 3 Report

Comments and Suggestions for Authors

It's better to remove the background line in the figure so that it looks more professional, and the abstract still needs to be improved 

Comments on the Quality of English Language

typo and other mistakes need to improved

Author Response

Dear Reviewer,

Thank you for your comments and suggestions, which will improve the manuscript. We have made changes in accordance with your requests.

R3: It would be better to remove the background line in the figure to make it look more professional, and the abstract still needs to be improved.

  • We thank the reviewer for his pertinent observations.
  • We have removed major gridlines from figures 2 and 3.
  • We have rewritten the abstract and added it in the comments to make it understandable and to fit relatively within the word count imposed by the Plants journal regulations.
